# QueryPose: Sparse Multi-Person Pose Regression via Spatial-Aware Part-Level Query

**Yabo Xiao**
BUPT
Beijing, China
xiaoyabo@bupt.edu.cn

**Kai Su**
ByteDance Inc.
Shanghai, China
sukai@bytedance.com

**Xioajuan Wang** *
BUPT
Beijing, China
wj2718@bupt.edu.cn

**Dongdong Yu**
OPPO Research Institute
Beijing, China
yudongdong@oppo.com

**Lei Jin**
BUPT
Beijing, China
jinlei@bupt.edu.cn

**Mingshu He**
BUPT
Beijing, China
hemingshu@bupt.edu.cn

**Zehuan Yuan** *
ByteDance Inc.
Beijing, China
yuanzehuan@bytedance.com

## Abstract

We propose a sparse end-to-end multi-person pose regression framework, termed QueryPose, which can directly predict multi-person keypoint sequences from the input image. The existing end-to-end methods rely on dense representations to preserve the spatial detail and structure for precise keypoint localization. However, the dense paradigm introduces complex and redundant post-processes during inference. In our framework, each human instance is encoded by several learnable spatial-aware part-level queries associated with an instance-level query. First, we propose the Spatial Part Embedding Generation Module (SPEGM) that considers the local spatial attention mechanism to generate several spatial-sensitive part embeddings, which contain spatial details and structural information for enhancing the part-level queries. Second, we introduce the Selective Iteration Module (SIM) to adaptively update the sparse part-level queries via the generated spatial-sensitive part embeddings stage-by-stage. Based on the two proposed modules, the part-level queries are able to fully encode the spatial details and structural information for precise keypoint regression. With the bipartite matching, QueryPose avoids the hand-designed post-processes and surpasses the existing dense end-to-end methods with 73.6 AP on MS COCO mini-val set and 72.7 AP on CrowdPose test set. Code is available at https://github.com/buptxyb666/QueryPose.

## 1 Introduction

Multi-person pose estimation (MPPE) aims to locate all person keypoints from the input image, which is a fundamental yet challenging task in computer vision. With the prevalence of deep learning techniques [1, 2, 3], MPPE has achieved remarkable progress and played an important role in many other vision tasks, such as activity recognition [4, 5, 6, 7] and pose tracking [8, 9].

---

*Corresponding author.

36th Conference on Neural Information Processing Systems (NeurIPS 2022).

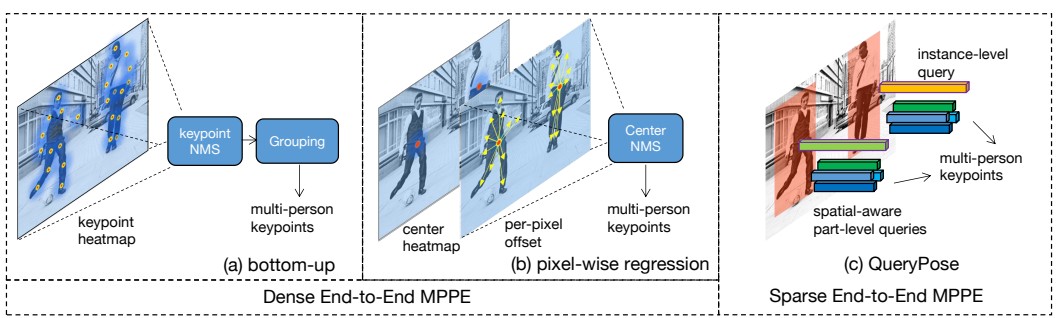

Figure 1: Comparison with the dense end-to-end solutions, including (a) bottom-up and (b) pixel-wise regression paradigms. (c) QueryPose is a sparse end-to-end solution.

In general, the existing multi-person pose estimation solutions can be summarized as top-down and bottom-up paradigms. The top-down strategy [10, 11, 12, 13, 14, 15] uses the human detector to predict person boxes and then leverages the single-person pose estimation model to localize the keypoints on each cropped human image. The two independent models lead to the non-end-to-end pipeline, or called two-stage pipeline. Moreover, the human detector involves extra memory as well as computational cost. The bottom-up strategy [16, 17, 18, 19] uses the keypoint heatmap to locate all person keypoints at first and then assigns them to individuals via heuristic grouping process, as shown in Figure 1(a). Some researches attempt to leverage densely pixel-wise regression [20, 21, 22, 23, 24, 25], as shown in Figure 1(b), which predicts the center heatmap and pixel-wise keypoint offset in parallel. It can be regarded as a special case of the bottom-up paradigm. Both of them can achieve end-to-end optimization thus are more efficient than the two-stage pipeline. However, in order to maintain the local details and spatial information, ones leverage the dense representations (e.g., keypoint or center heatmap). The dense manner requires the hand-crafted post-processes to suppress duplicate predictions or perform keypoint grouping during inference. The post-processes are non-differentiable and always involve many hand-designed parameters to tune. In this paper, we aim to build a sparse end-to-end MPPE solution to eliminate the complex and redundant post-processes.

Recently, query-based paradigm [26, 27, 28, 29, 30] has attracted much attention due to its purely sparse and end-to-end differentiable training/inference pipeline. The variants [30, 31, 29] are employed for several computer vision tasks and achieve promising performance. The success inspires us to construct a sparse end-to-end MPPE method via the learnable query. Nevertheless, how to effectively apply the query-based paradigm to the MPPE task is rarely explored. Intuitively, keypoints are similar to the corner points of the bounding box, and the instance-level object query can be used to regress the keypoint coordinates. However, we observe that it only achieves inferior performance. We argue that the instance-level query loses the local details and destroys the spatial structure, which are both critical for pose estimation task.

Towards the aforementioned issue, we introduce the learnable part-level queries to learn spatial-aware features and further construct a sparse end-to-end multi-person pose regression framework, termed QueryPose. First, we divide the human region into several local parts and present the Spatial Part Embedding Generation Module (SPEGM), which uses the local spatial attention mechanism to generate the spatial-sensitive part embeddings for enhancing part-level queries. Second, we introduce the Selective Iteration Module (SIM) to adaptively update the part-level queries via the generated spatial-sensitive part embeddings stage-by-stage. The local details and structural features in the learnable part-level queries will be enhanced, and the noise will be filtered. Based on the bipartite matching, we eliminate all hand-crafted post-processes and directly output the multi-person keypoint coordinate sequences for the input image. Without bells and whistles, our sparse framework outperforms all dense end-to-end methods on MS COCO [32] and CrowdPose [33].

The main contributions can be summarized into three aspects as follows:

- We propose to leverage the sparse learnable spatial-aware part queries to encode local spatial features. Specifically, we present the Spatial Part Embedding Generation Module (SPEGM) to generate the spatial-sensitive part embeddings via the local spatial attention mechanism, which preserves the local spatial features for enhancing the part-level queries.

- We further introduce the Selective Iteration Module (SIM) to adaptively update the learnable part-level queries by using the newly generated spatial-sensitive part embedding stage-by-stage. Accordingly, the encoded local details and spatial structure are strengthened, and the distractive information is ignored.

- With two proposed modules and one-to-one bipartite matching, we construct a purely sparse query-based MPPE framework, termed QueryPose, which removes the complex post-processes and directly outputs the multi-person keypoint coordinates. QueryPose achieves state-of-the-art performance and surpasses the most existing end-to-end MPPE methods on both MS COCO and CrowdPose.

## 2 Related Work

In this section, we review three relevant parts to our method including top-down paradigm, bottom-up paradigm and query-based paradigm.

**Top-down paradigm.** The top-down paradigm conducts single-person pose estimation on each cropped human image. Most top-down methods [13, 12, 10, 8] concentrate on learning the reliable high-quality feature representations for predicting keypoint heatmap. A few of researches [34, 35] try to improve the post-process and some others [36, 37] attempt to exploit regression methods to bypass the keypoint heatmap. However, top-down methods consist of the independent human detector and single-person pose estimation model, leading to the non-end-to-end optimization pipeline.

**Bottom-up paradigm.** The bottom-up paradigm [38, 18, 39, 16, 40] formulates this task as per-pixel keypoint positioning and grouping process. Moreover, densely pixel-wise regression [20, 21, 22, 23, 24, 41, 42] can be considered a special case of the bottom-up paradigm. One decomposes the MPPE into pixel-wise center localization and keypoint offset regression. However, both of them leverage the dense representations, thus introducing the non-differentiable post-processes during inference.

**Query-based paradigm.** The query-based paradigm [27, 28, 26, 29, 31, 30, 14] uses the learnable object query and one-to-one label assignment to achieve the purely sparse pipeline, which is applied to several vision tasks (e.g., object detection [27, 28, 26] and segmentation [29, 31, 30]). However, rare research employs the query-based paradigm for MPPE in end-to-end manner. PRTR [14] trivially uses the keypoint query with multi-head self-attention to probe the image feature, while losing the spatial local structural information, resulting in inferior performance. In our framework, we propose to leverage several spatial-aware part-level queries to learn local details and spatial structures.

## 3 Method

### 3.1 Overall Architecture

The overall framework is shown in Figure 2. The network is divided into three parts, including backbone, box decoder and keypoint decoder. The keypoint decoder consists of the proposed Spatial Part Embedding Generation Module and Selective Information Module. The box decoder and keypoint decoder are repeated $S$ times to build the cascade framework. $S$ is set to 6 by default, following previous methods [27, 28]. We only take stage $s$ as example for illustration.

**Backbone.** Given an input image $I$, we extract the multiple level features $P_2 \sim P_5$ via the feature pyramid network built upon the backbone, where $P_l$ indicates the feature with $1 / 2^l$ resolution of input and 256 channels. We feed multi-level features $P_2 \sim P_5$ to box decoder and only high-resolution $P_2$ to keypoint decoder.

**Box decoder.** The design follows Sparse R-CNN [28], we adopt a small set of learnable proposal boxes $B \in \mathbb{R}^{N \times 4}$ as dynamic anchors to provide the prior information of object position. Each proposal box corresponds to a learnable instance-level query. Concretely, we utilize the $7 \times 7$ RoIAlign operation to extract features $ROI_{box}$ according to the learnable boxes $B$. The extract features are one-to-one correspondence with learnable instance-level queries $Q_I \in \mathbb{R}^{N \times d}$. After exploring the self-attention across $Q_I$ by multi-head self-attention (MHSA) layer, we aggregate the extracted feature $ROI_{box}$ into instance-level query via the dynamic channel MLP (**DyMLP**$_{\textbf{channel}}$ in Eq. 1), and feed the enhanced instance-level query into regression head ( **Head**$_{\textbf{box}}$ in Eq. 1) to predict the

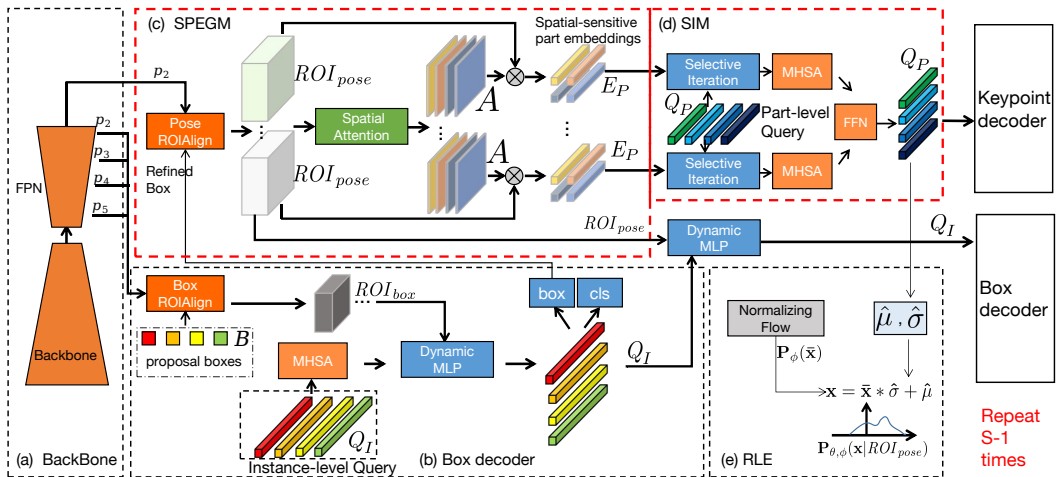

Figure 2: Overview of QueryPose. (a) Backbone for extracting multi-level features. (b) Box decoder. (c) Spatial Part Embedding Generation Module (SPEGM) employs the local spatial attention mechanism to produce spatial-sensitive part embeddings $E_P$. (d) Selective Iteration Module (SIM) adaptively updates the learnable part-level queries $Q_P$ by using the spatial-sensitive part embeddings. (e) The spatial-aware part-level quires are used to regress the keypoint coordinates via residual log-likelihood estimation(RLE) [37]. The box decoder is following Sparse RCNN. The keypoint decoder is composed of the proposed SPEGM and SIM.

offset for refining the input proposal box. The above process is summarized as follows:

$$ROI^s_{box} = \textbf{RoIAlign}(P_2 \sim P_5, \ B^{s-1}), \quad Q^s_I = \textbf{MHSA}(Q^{s-1}_I),$$
$$Q^s_I = \textbf{DyMLP}_{\textbf{channel}}(Q^s_I, \ ROI^s_{box}), \quad B^s = \Delta(\textbf{Head}_{\textbf{box}}(Q^s_I), \ B^{s-1}). \tag{1}$$

**Keypoint decoder.** After the above process, the instance-level query $Q_I$ has encoded the information of the human instance, thus can be used to regress the keypoints. However, it only obtains unsatisfactory performance. We argue that the keypoint localization requires more inherent spatial structure and local details than box regression, while $Q_I$ loses both spatial structure and local details.

In the light of this, we present to utilize several learnable spatial-aware part-level queries $Q_P$ to encode the human pose with spatial structure and local details maintained. $Q_P = \{Q_{P_n}|Q_{P_n} \in \mathbb{R}^{M \times d_p}\}^N_{n=1}$, where $n$ refers to $n$-th instance and $M$ is the number of part-level query for each instance. First, by using the refined boxes, we perform the $14 \times 14$ RoIAlign operation only on high-resolution feature $P_2$ to extract the more detailed features $ROI_{pose}$, which are more suitable for pose estimation. Second, the proposed Spatial Part Embedding Generation Module (SPEGM) is applied for producing the spatial-sensitive part embeddings $\textbf{E}_P = \{E_{P_n}|E_{P_n} \in \mathbb{R}^{M \times d_p}\}^N_{n=1}$ via the local spatial attention mechanism. Each spatial-sensitive part embedding focuses on the specific local part. Thus the local spatial details and structural information can be preserved. Third, we utilize the Selective Iteration Module (SIM) to adaptively update the learnable spatial-aware part-level queries via the spatial-sensitive part embeddings generated in the current stage. Consequently, the local details and spatial structure information of part-level queries are enhanced, and the noise is filtered. Then, the spatial-aware part queries are sent into the multi-head self-attention (denoted as MHSA) layer to explore the spatial relationships across the different local parts. Accordingly, the global structure of each person instance is also encoded into the part-level queries. Finally, we leverage the non-shared linear layers (denoted as Linear in Eq. 2) to regress the keypoint coordinates respectively. The computing procedure of keypoint decoder can be formulated as follows:

$$ROI^s_{pose} = \textbf{RoIAlign}(P_2, B^s), \quad E^s_P = \textbf{SPEGM}(ROI^s_{pose}),$$
$$Q^s_P = \textbf{SIM}(Q^{s-1}_P, E^s_P), \quad Q^s_P = \textbf{MHSA}(Q^s_P), \quad Pose^s = \textbf{Linear}(Q^s_P). \tag{2}$$

As shown in Figure 3(b), the instance-level query $Q_I$ is iterated across the box decoder and keypoint decoder serially, and the part-level queries $Q_p$ are iterated across keypoint decoder. The box decoder

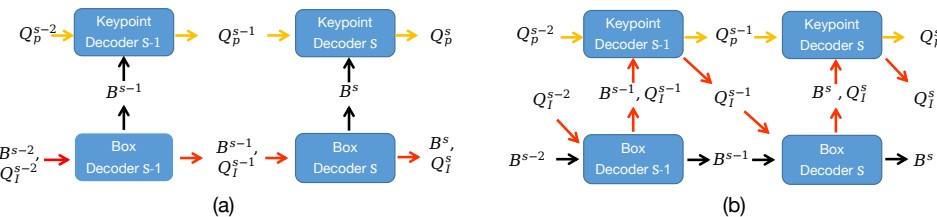

Figure 3: The iteration pipeline of the instance-level query $Q_I$ and part-level queries $Q_p$. (a) The instance-level query $Q_I$ is only iterated across the box decoder. (b) The instance-level query $Q_I$ is iterated across the box decoder and keypoint decoder serially.

of next stage adopts the predicted box of current stage as proposal box. The keypoint coordinates are predicted independently for each stage.

### 3.2 Spatial Part Embedding Generation Module

Considering that the instance-level query $Q_I$ loses the local details and spatial structure information compared with dense representations, we propose the Spatial Part Embedding Generation Module to generate the spatial-sensitive part embeddings $E_p$, which are able to maintain more spatial local features conducive to keypoint localization. The spatial-sensitive part embeddings $E_p$ are used to enhance the learnable spatial-aware part-level queries.

Given the single ROI feature $ROI_{pose} \in \mathbb{R}^{d \times H \times W}$ as example, We firstly use the stacked $3 \times 3$ convolutional layer and a deconvolutional layer to recover the higher resolution representation $ROI_{pose} \in \mathbb{R}^{d \times 2H \times 2W}$. Afterward, we take the spatial attention into consideration on the extracted ROI features $ROI_{pose}$. In particular, we utilize a $3 \times 3$ convolutional layer to squeeze the features along the channel dimension and produce the $M$-channels attention map $A \in \mathbb{R}^{M \times 2H \times 2W}$. Each channel corresponds to a spatial attention map for specific local parts, $M$ refers to the number of local parts. We flatten the attention map $A$ along the spatial dimension and normalize the sum of attention weight to 1.0 via the softmax function. Then, by manipulating the spatial attention maps of the specific local parts, we assemble the spatial feature to generate $M$ spatial-sensitive part embeddings. The process is formulated as: $E_P = \textbf{Linear}(A \otimes ROI_{pose}^T) \in \mathbb{R}^{M \times d_p}$, where $\otimes$ indicates dot product. The spatial-sensitive part embeddings focus on the different local parts, as shown in Figure 5. Thus the local details and structural information are sufficiently preserved compared with only a single instance-level query.

### 3.3 Selective Iteration Module

For enabling the learnable part-level queries $Q_P$ to learn the spatial-aware features, we propose the Spatial Part Embedding Generation Module (SPEGM) to generate several spatial-sensitive part embeddings $E_p$. However, we consider that due to the inaccurate bounding boxes and inconsistent optimization goals in the early stage, the spatial-sensitive part embeddings contain disturbance information. Motivated by this, we present the Selective Iteration Module to adaptively update and refine the part-level queries $Q_P$ via the spatial-sensitive part embeddings $E_p$.

First, we directly sum the learnable part-level queries $Q_P^{s-1} \in \mathbb{R}^{M \times d_p}$ output from previous stage and the spatial-sensitive part embeddings $E_P^s \in \mathbb{R}^{M \times d_p}$ generated in current stage. Second, we utilize a Multi-Layer Perceptron together with sigmoid operation to output two weight vectors $W_{E_P}^s$ and $W_{Q_P}^{s-1}$, which serve as two gates to control the contributions of the $E_P^s$ and $Q_P^{s-1}$. Finally, We leverage the weighted summation to fuse the $E_P^s$ and $Q_P^{s-1}$ via $W_{E_P}^s$ and $W_{Q_P}^{s-1}$. The above process is formulated as:

$$W_{E_P}^s, \ W_{Q_P}^{s-1} = \textbf{Sigmoid}(\textbf{MLP}(E_P^s + Q_P^{s-1})), \quad Q_P^s = W_{E_P}^s * E_P^s + W_{Q_P}^{s-1} * Q_P^{s-1}. \quad (3)$$

By using the Selective Iteration Module, the informative local details and spatial structure in part-level queries $Q_P$ will be refined, and the noise will be ignored.

## 3.4 Training and Testing Details

**Training.** In our framework, we leverage the instance-level query $Q_I$ to conduct the binary classification and bounding box regression, then use several spatial-aware part-level queries $Q_p$ to regress corresponding keypoints. The ground truth of each human instance contains the class label, box corners and keypoints coordinates, which are denoted as $C$, $\{(x_1^{box}, y_1^{box}), (x_2^{box}, y_2^{box})\}$, and $\{(x_k^{kp}, y_k^{kp})\}_{k=1}^K$ respectively. Following previous methods [27, 28], we adopt bipartite matching to perform one-to-one label assignment. The set-based loss function for instance-wise query is formulated as: $L_{Q_I} = \lambda_{cls} * L_{cls} + \lambda_{L1} * L_{L1} + \lambda_{giou} * L_{giou}$, where $L_{cls}$ refers to focal loss to perform binary classfication, $L_{L1}$ and $L_{giou}$ denote the L1 loss and generalized IoU loss between the predicted box and the corresponding label. the loss weight $\lambda_{cls}, \lambda_{L1}, \lambda_{giou}$ are set to 2.0, 5.0 and 2.0 respectively.

For keypoint regression, we leverage normalizing flow to capture the latent keypoint distribution $\mathbf{P}_{\theta,\phi}(\mathbf{x}|ROI_{pose})$, and model the keypoint regression loss from the perspective of maximum likelihood estimation following RLE [37]. The density distribution $\mathbf{P}_{\theta,\phi}(\mathbf{x}|ROI_{pose})$ reflects the probability of the keypoint annotation at the position $\mathbf{x}$ in $ROI_{pose}$, where $\theta$ and $\phi$ are the trainable parameters of our regression network and flow model respectively. For easier optimization, the flow model $F_\phi$ is used to map the simple distribution to deformed one $\mathbf{P}_\phi(\bar{\mathbf{x}})$. Our regression network outputs two values $\hat{\mu}$ and $\hat{\sigma}$ for shifting and rescaling the distribution $\mathbf{P}_\phi(\bar{\mathbf{x}})$, which is formulated as $\mathbf{x} = \bar{\mathbf{x}} * \hat{\sigma} + \hat{\mu}$. The process enables the network to learn how the output deviates from annotations.

Table 1: Ablation studies for exploring the query type and feature interaction method. IQ and PQ indicate the instance-level query and part-level query respectively.

| Query | Interaction | $AP$ | $AP_M$ | $AP_L$ |
|---|---|---|---|---|
| IQ | MHSA | 60.7 | 56.4 | 68.1 |
| PQ | MHSA | 62.6 | 58.2 | 69.9 |
| IQ | DyMLP**channel** | 60.4 | 56.3 | 67.4 |
| PQ | DyMLP**channel** | 63.0 | 58.7 | 70.4 |
| IQ | DyMLP**spatial** | 60.5 | 56.5 | 67.5 |
| PQ | DyMLP**spatial** | 62.8 | 58.6 | 70.3 |
| IQ | Spatial Attn. | 60.2 | 56.0 | 67.1 |
| PQ | SPEGM | 63.8 | 59.4 | 71.3 |

The different spatial-aware part queries are sent into the non-shared fully-connected layers to regress $\hat{\mu} \in \mathbb{R}^{K \times 2}$ and $\hat{\sigma} \in \mathbb{R}^{K \times 2}$, where $\hat{\mu}$ is the corresponding prediction of $\mu_g = \{(x_k^{kp}, y_k^{kp})\}_{k=1}^K$. Based on the matched pairs, the set-based keypoint regression loss for the spatial-aware part-level queries is formulated as:

$$L_{Q_P} = -\log \mathbf{P}_{\theta,\phi}(\mathbf{x}|ROI_{pose})|_{\mathbf{x}=\mu_g}. \tag{4}$$

It is noteworthy that $\mu_g$ is normalized by the height and width of the predicted bounding box and shifted to (-0.5, 0.5) for stabilizing the training process. The keypoint out of the predicted bounding box will be masked. The total loss of our framework can be written as: $L_{total} = L_{Q_I} + L_{Q_P}$.

**Testing.** $\hat{\sigma}$ can be regarded as the uncertainty score of keypoint localization, thus we define the final pose score as $\frac{1}{K}\sum_{k=1}^K (1 - \hat{\sigma}_k) * \bar{C}$, where $\bar{C}$ is the predicted instance score. During inference, our method eliminates hand-crafted post-procedures (e.g., keypoint NMS, center NMS and grouping) and only need to select the pose candidates with high pose score.

## 4 Experiments and Analysis

In this section, we briefly illustrate the dataset, evaluation metric as well as implementation details in subsection 4.1. Next, we delve into the proposed modules and conduct the comprehensive ablation experiments to verify the effectiveness in subsection 4.2. Finally, we compare QueryPose with the previous methods on both MS COCO and CrowdPose in subsection 4.3.

### 4.1 Experimental Setup

**Dataset.** MS COCO [32] is a large-scale 2D human pose benchmark, which is split into train2017, mini-val, test-dev2017 sets. We train our model on COCO train2017 set with 57k images, conduct the evaluation on the mini-val set with 5k images and test-dev2017 set with 20K images respectively. CrowdPose [33] contains 20,000 images with 80,000 annotated human instances. The train, validation and test set are partitioned in proportional to 5:1:4.

Table 2: Ablation experiments.

(a) Ablative studies for the iteration methods of learnable part-level queries.

| Iteration | $AP$ | $AP_M$ | $AP_L$ |
|---|---|---|---|
| None | 63.8 | 59.4 | 71.3 |
| Summation | 64.2 | 60.1 | 71.7 |
| Concatenation | 64.1 | 59.6 | 71.6 |
| Selective Iter. | 64.9 | 60.4 | 72.3 |

(b) The contributions of two proposed modules in overall framework.

| Method | $AP$ | $AP_M$ | $AP_L$ |
|---|---|---|---|
| Baseline | 59.2 | 55.1 | 66.0 |
| + SPEGM | 63.8 | 59.4 | 71.3 |
| + SIM | 64.9 | 60.4 | 72.3 |

(c) The influence for the dimension of spatial-sensitive part-level query.

| Dimension | $AP$ | $AP_M$ | $AP_L$ |
|---|---|---|---|
| 32 | 64.1 | 59.8 | 71.3 |
| 64 | 64.4 | 60.0 | 71.8 |
| 128 | 64.9 | 60.4 | 72.3 |
| 256 | 64.7 | 60.5 | 72.1 |

(d) Ablative studies for the different part division schemes. Scheme (a), (b), (c), (d) are corresponding to the Figure 4 (a), (b), (c), (d).

| Scheme | Part | $AP$ | $AP_M$ | $AP_L$ |
|---|---|---|---|---|
| (a) | 17 | 64.7 | 60.4 | 72.1 |
| (b) | 13 | 64.6 | 60.1 | 72.0 |
| (c) | 7 | 64.9 | 60.4 | 72.3 |
| (d) | 5 | 64.5 | 60.5 | 71.7 |

**Evaluation Metric.** We leverage average precision and average recall based on different Object Keypoint Similarity (OKS) [32] thresholds to evaluate performance. For COCO dataset, $AP_M$ and $AP_L$ indicate AP over medium and large-sized instances respectively. For CrowdPose, $AP_E$, $AP_M$, $AP_H$ refer to AP scores over easy, medium and hard instances according to dataset annotations.

**Implementation Details.** During training, we use random horizontal flip and random crop with probability 0.5 to augment the training samples. The input images are randomly resized, and the short side is randomly sampled from 480 to 800 pixels with the aspect ratio kept. The number of the sparse instance-level query is set to 100 by default, it can even be reduced to 50 without significant performance degradation due to the foreground only containing the person. The model is trained by using AdamW [43] optimizer with a mini-batch size of 16 (2 per GPU) on eight Tesla A100 GPUs. The initial learning rate is linearly scaled to 2.5e-5 and dropped $10\times$ after 220k, 260k iterations, and terminated at 280k iterations ($2\times$ training schedule). The learning rate of the flow model is $10\times$ of the regression model. All codes are implemented based on Detectron2 [44]. During inference, we keep the aspect ratio and resize the short side of the images to 800 pixels.

## 4.2 Ablation Study

We conduct the ablative studies on MS COCO mini-val set to explore the contributions of the proposed components and delve into the superiority of its design. All experiments adopt ResNet-50-FPN and $1\times$ training scheme. We first define the baseline as employing the stacked $3\times3$ conv-relu + global average pooling on $ROI_{pose}$ to directly regress keypoint coordinates.

**Analysis of the part-level query.** As shown in Table 1, to verify the superiority of the part-level query over the instance-level query, we leverage four different interaction methods with image feature $ROI_{pose}$: (1) MHSA refers to multi-head self-attention layer. (2) DyMLP**channel** indicates the dynamic MLP acting on the channel dimension, where the weight of MLP is generated by the corresponding query. (3) DyMLP**spatial** refers to the dynamic MLP acting on the spatial dimension. (4) Spatial Attn. denotes using global spatial attention to generate the instance embedding to enhance the instance-level query. We observe that part-level query can consistently achieve the notable improvements than instance-level query in case of utilizing the different ways to interact with the image features $ROI_{pose}$. This phenomenon proves that part-level query is able to preserve more details of local area than instance-level query.

Table 3: Ablation studies for the iteration pipeline of the instance-level query.

| Iter. pipeline | $AP^{box}$ | $AP^{kps}$ |
|---|---|---|
| Figure 3(a) | 55.3 | 64.1 |
| Figure 3(b) | 56.9 | 64.9 |

**Analysis of Spatial Part Embedding Generation Module.** To study the superiority of the Spatial Part Embedding Generation Module (SPEGM), we compare the performance with different interaction methods in Table 1. Notably, using the SPEGM to generate spatial-sensitive part embeddings for boosting the learnable part-level queries achieves the superior performance (63.8 AP) than the other interaction ways. Furthermore, SPEGM improves the baseline by 4.6 AP as reported in Table 2(b). We suggest that the local spatial attention is able to provide the spatial prior information and focus on

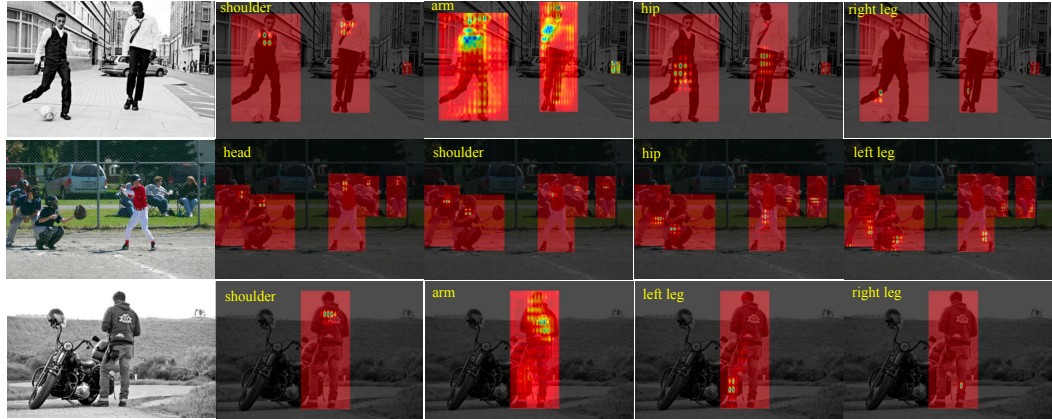

Figure 5: The visualizations of the local spatial attention maps $A$, which correspond to the local parts of each person. The local salient region is visualized by bright color. The backgrounds of person regions are rendered to red. Best viewed after zooming in.

the informative local regions, as shown in Figure 5. Furthermore, compared with dynamic MLP with over 10M parameters, SPEGM is more effective and efficient with only 1.1M parameters.

**Analysis of Selective Iteration Module.** Selective Iteration Module (SIM) is presented to adaptively update the learnable spatial-aware part queries output from the previous stage. We conduct the controlled experiments to investigate the superiority of the proposed SIM compared with the other two iteration methods, including summation and concatenation. As reported in Table 2(a), the Selective Iteration Module achieves 64.9 AP and improves the non-iteration structure by 1.1 AP. Moreover, SIM outperforms the summation and concatenation by 0.7 and 0.8 AP, respectively. We consider that the Selective Iteration Module can adaptively boost the informative spatial details and filter the noise.

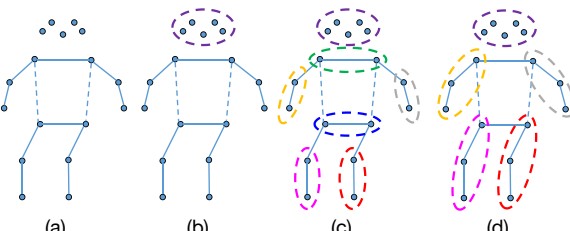

Figure 4: The different part division schemes. (a) Each keypoint is corresponding to a part-level query. (b) All keypoints on head are corresponding to a part-level query. (c) Each divided part contains rigid structure and corresponds to a part-level query. (d) The head and four limbs are corresponding to a part-level query respectively.

**Analysis of the dimension for the part-level query.** The part-level queries are used to encode the local spatial detail and structure. Based on the overall network, we explore the dimension of the spatial-aware part query. As reported in Table 2(c), we observed that setting the dimension to 128 is sufficient, and the larger dimension can not bring additional gains.

**Analysis of the part division scheme.** As shown in Figure 4 and Table 2(d), we probe four different part division schemes to design an efficient and effective partition strategy. The scheme (c) achieves the optimal performance. Compared with the scheme (a) and (b), the part-level query in scheme (c) is capable of obtaining local structural information of adjacent keypoints. Moreover, each part in scheme (c) is rigid without articulated structure, while the part in scheme (d) is deformable, thus obtaining a slightly worse result.

**Iteration pipeline of the instance-level query.** As shown in Figure 3 and Table 3, we explore the two different iteration pipeline of the instance-level query $Q_I$. We observe that the instance-level query iterated across box decoder and keypoint decoder serially can improve 1.6 $AP^{box}$ and 0.8 $AP^{kps}$ than only iterated across box decoder.

## 4.3 Results

**COCO Mini-val set.** Table 4 reports the results of end-to-end methods on COCO mini-val set. Compared with dense end-to-end methods, QueryPose with ResNet-50 outperforms Mask R-CNN

Table 4: Comparisons with the end-to-end MPPE methods on the COCO mini-val set. Note that all results are reported for single-scale testing. *Flip* refers to using the flip testing to boost performance.

| Methods | Backbone | $AP$ | $AP_{50}$ | $AP_{75}$ | $AP_M$ | $AP_L$ | Time [ms] |
|---|---|---|---|---|---|---|---|
| Dense End-to-End MPPE | | | | | | | |
| Mask R-CNN [45] | ResNet101 | 66.1 | 87.7 | 71.7 | 60.5 | 75.0 | 128 |
| HrHRNet-W48 [16] | HrHRNet-W48 | 66.6 | 85.3 | 72.8 | 61.7 | 74.4 | 233 |
| HrHRNet-W48 (*Flip*) [16] | HrHRNet-W48 | 69.8 | 87.2 | 76.1 | 65.4 | 76.4 | 317 |
| SWAHR-W48 [40] | HrHRNet-W48 | 67.3 | 87.1 | 72.9 | 62.1 | 75.0 | 242 |
| SWAHR-W48 (*Flip*) [40] | HrHRNet-W48 | 70.8 | 88.5 | 76.8 | 66.3 | 77.4 | 339 |
| CenterGroup-W48 [46] | HrHRNet-W48 | 69.1 | - | - | - | - | - |
| CenterNet-HG (*Flip*) [21] | HG-104 | 64.0 | - | - | - | - | 135 |
| Mask R-CNN + RLE [37] | ResNet101 | 66.7 | 86.7 | 72.6 | - | - | - |
| DEKR-W48 [24] | HRNet-W48 | 67.1 | 87.7 | 73.9 | 61.5 | 77.1 | 197 |
| DEKR-W48 (*Flip*) [24] | HRNet-W48 | 71.0 | 88.3 | 77.4 | 66.7 | 78.5 | 284 |
| AdaptivePose [20] | HRNet-W48 | 70.0 | 87.5 | 76.1 | 65.4 | 77.1 | 110 |
| LQR (*Flip*) [41] | HRNet-W48 | 72.4 | 89.1 | 79.0 | 67.3 | 80.4 | 297 |
| Sparse End-to-End MPPE | | | | | | | |
| PRTR [14] | HRNet-W48 | 66.2 | 85.9 | 72.1 | 61.3 | 74.4 | - |
| QueryPose | ResNet50 | 68.7 | 88.6 | 74.4 | 63.8 | 76.5 | 70 |
| QueryPose | Swin-L | 73.3 | **91.3** | 79.5 | 68.5 | **81.2** | 117 |
| QueryPose | HRNet-W32 | 72.4 | 89.8 | 78.6 | 67.9 | 79.7 | 100 |
| QueryPose | HRNet-W48 | **73.6** | 90.3 | **79.7** | **69.3** | 80.8 | 105 |

Table 5: Comprehensive comparisons on COCO test-dev set. $*$ indicates the refinement by a well-trained single-person pose estimation model. $\dagger$ refers to test-time augmentation (e.g., flip or multi-scale testing).

| Methods | $AP$ | $AP_{50}$ | $AP_{75}$ | $AP_M$ | $AP_L$ |
|---|---|---|---|---|---|
| Two-Stage Methods | | | | | |
| G-RMI [17] | 64.9 | 85.5 | 71.3 | 62.3 | 70.0 |
| Integral Pose [36] | 67.8 | 88.2 | 74.8 | 63.9 | 74.0 |
| CPN$^\dagger$ [10] | 72.1 | 91.4 | 80.0 | 68.7 | 77.2 |
| SimpleBaseline$^\dagger$ [8] | 73.7 | 91.9 | 81.1 | 70.3 | 80.0 |
| HRNet$^\dagger$ [13] | **75.5** | **92.5** | **83.3** | **71.9** | **81.5** |
| End-to-End Methods | | | | | |
| CMU-Pose$^{*\dagger}$ [39] | 61.8 | 84.9 | 67.5 | 57.1 | 68.2 |
| AE$^{*\dagger}$ [19] | 65.5 | 86.8 | 72.3 | 60.6 | 72.6 |
| CenterNet-HG [21] | 63.0 | 86.8 | 69.6 | 58.9 | 70.4 |
| SPM $^{*\dagger}$ [22] | 66.9 | 88.5 | 72.9 | 62.6 | 73.1 |
| PointSetNet$^\dagger$ [42] | 68.7 | 89.9 | 76.3 | 64.8 | 75.3 |
| HrHRNet$^\dagger$ [16] | 70.5 | 89.3 | 77.2 | 66.6 | 75.8 |
| DEKR$^\dagger$ [24] | 71.0 | 89.2 | 78.0 | 67.1 | 76.9 |
| CenterGroup$^\dagger$ [46] | 71.1 | 90.5 | 77.5 | 66.9 | 76.7 |
| LQR$^\dagger$ [41] | 71.7 | 90.4 | 78.7 | 67.3 | 78.5 |
| AdaptivePose$^\dagger$ [20] | 71.3 | 90.0 | 78.3 | 67.1 | 77.2 |
| QueryPose-W48 | **72.3** | 91.5 | 78.7 | **67.8** | 79.0 |
| QueryPose-SwinL | 72.2 | **92.0** | **78.8** | 67.3 | **79.4** |

[45] with ResNet-101 by 2.6 AP. By only using HRNet-W32, we achieve 72.4 AP and markedly outperform all dense counterparts with larger backbone. QueryPose with HRNet-W48 obtains 73.6 AP and significantly surpasses the previous most competitive HrHRNet-W48 (*Flip*) [16], SWAHR-W48 (*Flip*) [40], CenterGroup-W48[46] by 3.8 AP, 2.8 AP, 4.5 AP respectively. Furthermore, QueryPose outperforms the regression-based AdaptivePose-W48 [20] and DEKR-W48(*Flip*) [24] by 3.6 AP and 2.6 AP. QueryPose obtains 7.4 AP improvements over the existing sparse end-to-end method PRTR [14]. Finally, we use the transformer-based backbone Swin-Large [47] and also achieve 73.3 AP, which proves that our method is compatible for various network architecture. QueryPose eliminates the time-consuming post-processes and achieves the faster inference speed compared with the above state-of-the-art dense competitors.

Table 6: Comparisons between the previous two-stage and end-to-end MPPE methods on CrowdPose test set. $^\dagger$ indicates multi-scale testing.

| Methods | $AP$ | $AP_{50}$ | $AP_{75}$ | $AP_E$ | $AP_M$ | $AP_H$ |
|---|---|---|---|---|---|---|
| Two-Stage MPPE | | | | | | |
| Rmpe [11] | 61.0 | 81.3 | 66.0 | 71.2 | 61.4 | 51.1 |
| SimpleBaseline [8] | 60.8 | 84.2 | 71.5 | 71.4 | 61.2 | 51.2 |
| CrowdPose [33] | 66.0 | 84.2 | 71.5 | 75.5 | 66.3 | 57.4 |
| End-to-End MPPE | | | | | | |
| CMU-Pose [39] | - | - | - | 62.7 | 48.7 | 32.3 |
| Mask-RCNN [45] | 57.2 | 83.5 | 60.3 | 69.4 | 57.9 | 45.8 |
| SWAHR-W48 [40] | 71.6 | 88.5 | 77.6 | 78.9 | 72.4 | 63.0 |
| DEKR-W48$^\dagger$ [24] | 68.0 | 85.5 | 73.4 | 76.6 | 68.8 | 58.4 |
| AdaptivePose-W48$^\dagger$ [20] | 69.2 | 87.3 | 75.0 | 76.7 | 70.0 | 60.9 |
| HigherHRNet-W48$^\dagger$ [16] | 67.6 | 87.4 | 72.6 | 75.8 | 68.1 | 58.9 |
| CenterGroup-W48$^\dagger$ [46] | 70.0 | 88.9 | 75.1 | 76.8 | 70.7 | 62.2 |
| QueryPose-W48 | 72.1 | 89.4 | 78.0 | **79.8** | 73.3 | 63.0 |
| QueryPose-SwinL | **72.7** | **91.7** | **78.1** | 79.5 | **73.4** | **65.4** |

**COCO test-dev2017 set.** As shown in Table 5, we achieve 72.3 AP with HRNet-W48 and 72.2 AP with Swin-Large without flip and multi-scale testing. QueryPose exceeds all end-to-end methods and narrows the gap with two-stage methods. The results verify the superior keypoint positioning ability of QueryPose in single-forward pass.

**CrowdPose test set.** We further evaluate the QueryPose on CrowdPose. The model is trained on the train and val set, then evaluated on test set as previous methods [16, 40, 24]. We list the comparisons in Table 6. QueryPose achieves 72.1 AP with HRNet-W48 and 72.7 AP with Swin-large on CrowdPose test set. The results outperform the most existing MPPE methods.

**Discussion.** We leverage the sparse spatial-aware part-level queries to encode the local detail and spatial structure instead of dense feature representation. Our sparse method achieves the better performance, proving that dense heatmap is not all you need for pose estimation. In crowded scenes, the sparse paradigm probably avoids the issues caused by the NMS process (e.g., the heavily overlapped instances or keypoints may be removed after center or keypoint NMS). Accordingly, the sparse paradigm may be more compatible for crowded scenes.

Furthermore, we observe that the typical single-stage regression-based methods (without ROI) require the online auxiliary keypoint heatmap learning during training stage, which can bring 1.5-2.5 AP improvements for different approaches (e.g., AdaptivePose[20], PointSetNet[42]). In contrast, The online auxiliary keypoint heatmap learning is useless for QueryPose, while the heatmap pretrain is more suitable for QueryPose. We only leverage HRNet to verify the phenomenons and bring the 1.5 AP improvements.

## 5 Conclusion

In this paper, we present a sparse end-to-end multi-person pose regression framework termed as QueryPose. With two proposed modules, i.e., Spatial Part Embedding Generation Module and Selective Iteration Module, QueryPose is capable of utilizing the sparse spatial-aware part-level queries to capture the local spatial features instead of previous dense representations and achieving the better performance than most dense MPPE methods. Our method proves the potential of the sparse regression method for the multi-person pose estimation task. We believe that our core insight is able to benefit some other methods and inspire future research.

## 6 Acknowledgements

This work is supported by the National Natural Science Foundation of China No.62071056, No.61871046 and No.62102039.

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
