# Supplementary Material for
# QueryPose: Sparse Multi-Person Pose Regression via Spatial-Aware Part-Level Query

## Abstract

In this supplementary material, we first present more results in Appendix A, including 1) more clear and intuitive visualizations of local attention map; 2) qualitative results on MS COCO [1] mini-val set; 3) qualitative results on CrowdPose [2] test set. Then, we give a broader discussion about our method in Appendix B.

## A   Results

**Qualitative results.** First, As shown in Figure 1, we present more visualizations of local attention maps for specific parts. We observe that the spatial attention can precisely focus on the corresponding local part. Moreover, in case of occluded or invisible keypoints, the corresponding spatial attention will concentrate on the region with more semantic context to infer the occluded keypoints. Second, we visualize the predicted multi-person pose on MS COCO, as shown in Figure 2. QueryPose can precisely estimate multi-person pose in various complex scenarios. Finally, we visualize the predicted multi-person pose on CrowdPose, as shown in Figure 3. QueryPose is robust for the crowded multi-person scenarios, including numerous occluded and overlapped human bodies, blurred and twisted limbs.

## B   Discussion

This paper discusses the difference between the dense and sparse representations in multi-person pose estimation task. We leverage the sparse spatial-aware part-level queries to encode the local detail and spatial structure instead of dense feature representation. Our sparse method achieves the better performance, proving that dense heatmap is not all you need for pose estimation. In crowded scenes, the sparse paradigm probably avoids the issues caused by the NMS process (e.g., the heavily overlapped instances or keypoints may be removed after center or keypoint NMS). Accordingly, the sparse paradigm may be more compatible for crowded scenes.

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

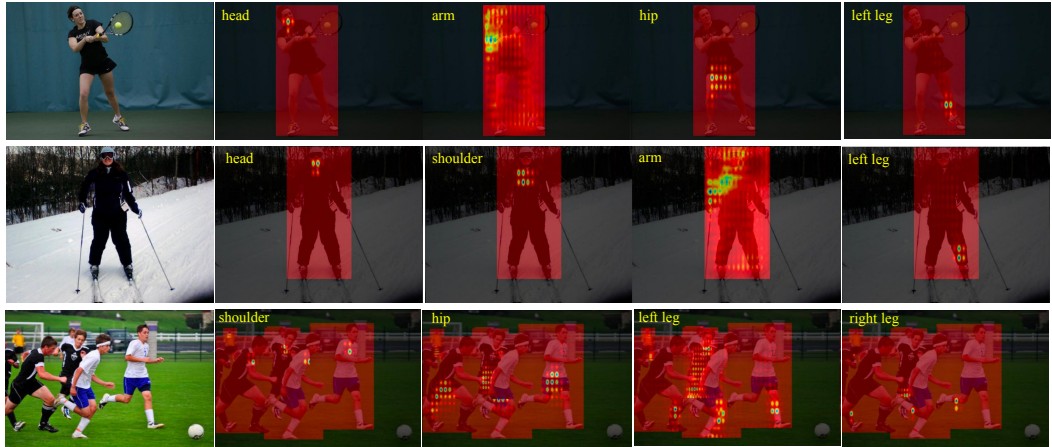

Figure 1: The visualizations of local spatial attention maps. Each local spatial attention focuses on the specific human part. The human backgrounds are visualized by red. The salient local regions are visualized by bright color.

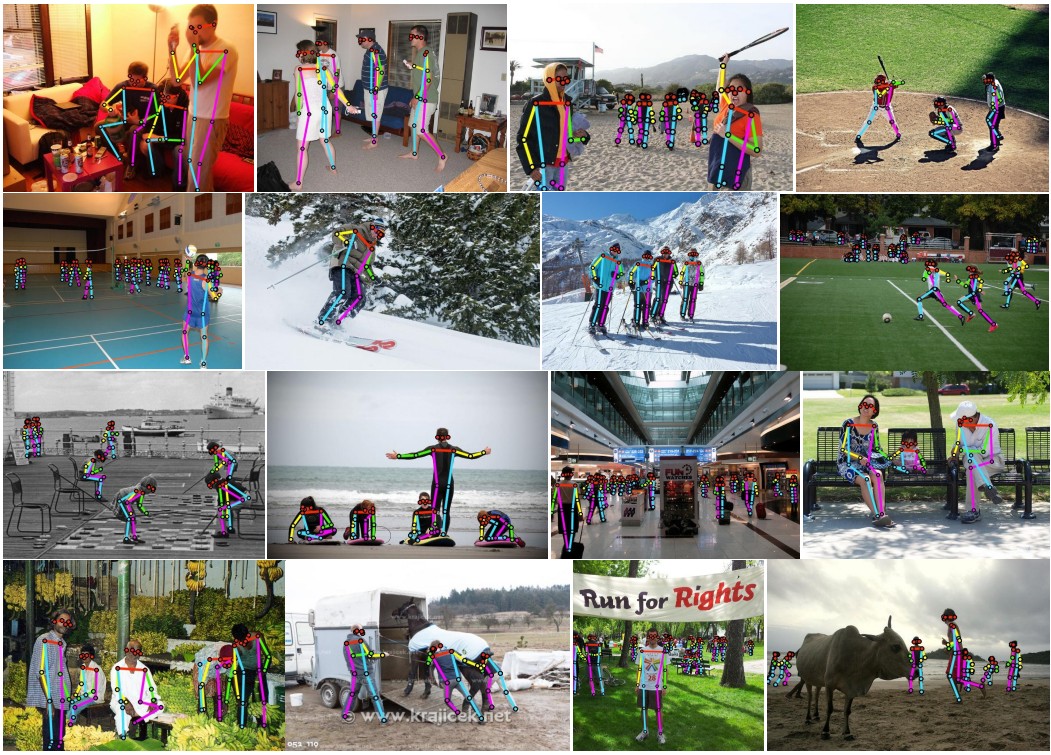

Figure 2: Examples of the predicted multi-person pose on MS COCO. QueryPose deals well with pose deformation and scale variation.

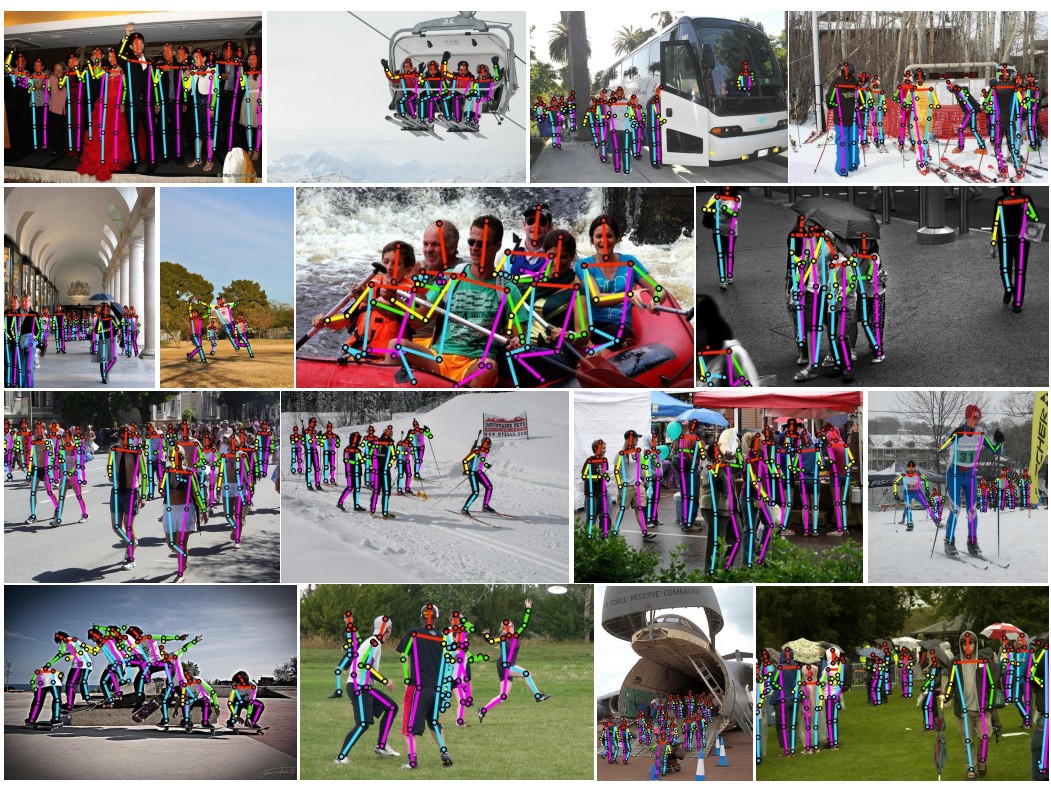

Figure 3: Examples of the predicted multi-person pose on CrowPose. QueryPose is robust for the challenging crowded multi-person scenarios, including occlusions, blurred and twisted limbs.