# OpenReview forum: "QueryPose: Sparse Multi-Person Pose Regression via Spatial-Aware Part-Level Query"
_NeurIPS.cc/2022/Conference — NeurIPS 2022 Accept_

### Official Review · Reviewer_ve12 · 2022-07-09

**Rating:** 6
**Confidence:** 4
**Soundness:** 3 good
**Presentation:** 3 good
**Contribution:** 3 good

**Summary:**

The paper proposes a new method for multiperson-pose estimation in an end-to-end learnable scheme. It brings together ideas from Mask-RCNN/Roi-Align and Transformers/Attention in a cascade scheme in which the box and keypoint predictions are refined in several stages. The paper contains ablation studies which examine the effectiveness of the different modules and reports strong results on the COCO keypoints and CrowdPose pose estimation benchmarks.

**Questions:**

I think the key missing information is how fast/slow the proposed method is during training and inference times.

The proposed system is quite complex and hard to reproduce given the information provided in the paper or supplementary material. However sharing the code as the authors intend to do based on the abstract will be sufficient to address this point.

**Limitations:**

Not very much applicable to this paper.

**Strengths And Weaknesses:**

Strengths:

+ Good experimental results on well-known competitive benchmarks.
+ Although the method is complex, the paper does a good job explaining what each of the several modules does. Figure 2 is particularly helpful.
+ Authors promise to share code. This is very helpful because the method is quite complex and it would be very hard to reproduce the reported results using the description provided in the paper.

Weaknesses:
- The proposed method is quite complex. There are several modules involved.
- No runtime information is provided. It is hard to tell how fast/slow the method is. Both training and inference times need to be provided in the updated manuscript.

Some minor typos:

Typo in Fig 2(e), Normalizing
Line 166: Multilayer Perceptron

---

> ### Author Response · Authors · 2022-08-02
> **Response to Reviewer ve12**
>
> Thank you for the careful reviews and constructive suggestions. We will clarify the questions as follows.
>
> **Q1: The proposed method is quite complex. There are several modules involved.**
>
> **A1**: Thanks for your concern. The whole framework can be divided into three parts, including backbone, box decoder and keypoint decoder.  The keypoint decoder consists of the proposed Spatial Part Embedding Generation Module(SPEGM) and Selective Information Module(SIM). SPEGM takes ROI feature as inputs, and leverages the local spatial attention to generate spatial-sensitive part embeddings for preserving local spatial information. SIM uses the spatial-sensitive part embeddings to adaptively update the part query. We concentrate on the keypoint decoder and build a sparse end-to-end MPPE pipeline via the sparse part queries.
>
> **Q2: No runtime information is provided. It is hard to tell how fast/slow the method is. Both training and inference times need to be provided in the updated manuscript.**
>
> **A2**: Thanks for your comments. We report the training and inference time of QueryPose with different backbone.
> Backbone | AP| Training Time [h]| Inference Time [ms]
> --|--|--|--
> ResNet50| 68.7| 42h| 70
> HRNet-W32|72.4|58h| 100
> HRNet-W48|73.6| 62h| 105
> Swin-L | 73.3| 67h| 117
>
> Compared with the representative dense competitors, we observe QueryPose is training-efficient with fewer training epochs . Furthermore, QueryPose eliminates time-consuming post-processes and achieves the faster speed. The detailed comparisons are reported as follows:
>
> Methods | AP| Training Epoch| Training Time [h]| Inference Time [ms]
> --|--|--|--|--
> HrHRNet-W48 [16]| 69.8 |300 | >140h | 317
> SWAHR-W48 [39]|70.8| 300 | >140h| 339
> DEKR-W48 [24] | 71.0|140 | >80h| 284
> AdaptivePose-W48 [20]| 70.0|280 | 140h| 110
> LQR-W48 [40]| 72.4| 170 | 100h| 297
> QueryPose-W48| 73.6| 68 | 62h| 105
>
> We will provide the training and inference times in the revision.
>
> **Q3: Some minor typos.**
>
> **A3**: Thanks, we revise our paper carefully in the revision.
>
> **Q4: The proposed system is quite complex and hard to reproduce given the information provided in the paper or supplementary material. However sharing the code as the authors intend to do based on the abstract will be sufficient to address this point.**
>
> **A4**: Thanks. We will release the code for the convenience of reproducing our method.

---

> > ### Author Response · Authors · 2022-08-09
> > **Looking forward to hearing the response of reviewer ve12**
> >
> > Thank you for the previous insightful comments. We also would like to receive your further response about our clarifications.

---

### Official Review · Reviewer_YZdd · 2022-07-10

**Rating:** 6
**Confidence:** 3
**Soundness:** 3 good
**Presentation:** 3 good
**Contribution:** 2 fair

**Summary:**

The paper presents a query-based human pose estimation approach. It builds upon Sparse-RCNN and design Spatial Part Embedding Generation Module (SPEGM) and Selective Iteration Module (SIM) to further improve the performance. Experiments on COCO and CrowdPose validates the effectiveness of the proposed method.


**Questions:**

See above.

**Ethics Review Area:**

["I don’t know"]

**Limitations:**

Yes

**Strengths And Weaknesses:**

Pros:
1.	The proposed method is interesting. It adopts the query-based detection framework for one-stage pose estimation and designs heads and interaction between instance query and pose query.
2.	It achieves the state-of-the-art performance among the one-stage approaches.

Cons:
1.	Two-stage vs one-stage, computational complexity
a)	From my point of view, the proposed method is not a typical “one-stage” approach. One major drawback of the two-stage top-down approaches is that their computational cost increases with the number of people (This is also pointed out by the authors in Introduction). But the proposed method (QueryPose) also suffers from such problem. It first locates the bounding box and then estimates the human poses for each bounding box. Isn’t the whole pipeline similar to Mask-RCNN? So I think it is more like a top-down approach, and its performance should be compared with other top-down approaches.
b)	The total computational complexity and runtime performance are concerning, but not reported in the paper.
2.	The paper writing can be improved. Especially, Figure2 is not clear.

---

> ### Author Response · Authors · 2022-08-02
> **Response to Reviewer YZdd**
>
> Thank you for the careful reviews and constructive suggestions. We clarify the questions as follows.
>
> **Q1: The proposed method is not a typical “one-stage” approach. One major drawback of the two-stage top-down approaches is that their computational cost increases with the number of people. But the proposed method (QueryPose) also suffers from such problem.**
>
> **A1**: Thanks for your concern. In this paper, we classify multi-person pose estimation (MPPE) methods from the perspective of two-stage and end-to-end optimization. We argue QueryPose is a sparse end-to-end MPPE framework, which can directly output multi-person keypoint sequences in parallel without non-differentiable post-processes.
>
> Two-stage top-down approaches use two independent models including a human detector and a single-person pose estimator. The raw image is cropped to several normalized images according to the detected person boxes as input of the single-person pose estimator. The input images increase with the number of people, thus computational costs are increased. QueryPose is a query-based method. As all known, the number of queries is set to a fixed and small number(e.g., 50 or 100) for both training and testing stages. All queries are calculated in parallel in box and keypoint decoders. Therefore, QueryPose does not suffer from such problem.
>
> **Q2: It first locates the bounding box and then estimates the human poses for each bounding box. Isn’t the whole pipeline similar to Mask-RCNN? So I think it is more like a top-down approach, its performance should be compared with other top-down approaches.**
>
> **A2**: To some extent, the pipelines of QueryPose and Mask RCNN are similar. Both of them take the multi-person image as input,  utilize an end-to-end network to locate the bounding box and estimate the human pose. However, Mask RCNN leverages the dense anchors and heatmap to perform MPPE, thus we classify it as dense end-to-end method in Table 4.
>
> In contrast to two-stage top-down approaches using two independent models to perform MPPE, QueryPose leverages an end-to-end network to directly predicts multi-person keypoint sequences from the raw image via sparse learnable queries. We list the comprehensive comparisons with existing dense/sparse end-to-end methods in Table 4. Furthermore, we also compared it with the two-stage methods in Tables 5 and 6. QueryPose not only outperforms all existing end-to-end methods but narrows the gap with the two-stage methods.
>
> **Q3: The total computational complexity and runtime performance are concerning, but not reported in the paper**
>
> **A3**:  We measure the inference time and FLOPs of QueryPose with ResNet50, HRNet-W32, HRNet-W48 and Swin-L.
> | Backbone     |  AP |  Time [ms] |  FLOPs  |
> |-------- |-------- |------ |------- |
> |ResNet50      | 68.7  |    70   | 153G |
> |HRNet-W32  | 72.4 |    100 | 237G    |
> |HRNet-W48  | 73.6 | 105 |     334G   |
> |Swin-L |   73.3 |    117 |      545G   |
>
> Compared with the state-of-the-art dense end-to-end methods, QueryPose eliminates time-consuming post-processes and achieves the faster speed. The inference time of QueryPose-R50 is 70ms with 153G FLOPs, which is faster than Mask RCNN-R50 (114ms) with 238G FlOPs.  With the same backbone, QueryPose outperforms the representative DEKR, HrHRNet, SWAHR, and LQR  in terms of accuracy and speed. More comparisons of runtime performance are reported in the revision.
> | Methods |  Backbone  | AP| Time [ms] |
> |---|-- |--|---|
> |HrHRNet [16] | HRNet-W48 |69.8 |317    |
> | SWAHR [39] |   HRNet-W48| 70.8 |     339  |
> |DEKR [24]   | HRNet-W48|71.0 | 284|
> |LQR [40]  | HRNet-W48|72.4| 297   |
> | QueryPose  |   HRNet-W48  |    73.6  | 105  |
>
> **Q4: The paper writing can be improved. Especially, Figure2 is not clear.**
>
>
> **A4**: Thanks, we will re-organize our paper writing and improve Figure 2 for clearer presentation.

---

> > ### Comment · Reviewer_YZdd · 2022-08-08
> > **Thanks for the response**
> >
> > Thanks. I have read the responses and all my concerns have been addressed.

---

> > > ### Author Response · Authors · 2022-08-09
> > > **Thanks again for Reviewer YZdd**
> > >
> > > We are glad to hear that the concerns have been addressed. Thanks again for the time and effort in reviewing our paper. The constructive suggestions help us make our paper better.

---

### Official Review · Reviewer_2Ux2 · 2022-07-11

**Rating:** 6
**Confidence:** 4
**Soundness:** 3 good
**Presentation:** 2 fair
**Contribution:** 3 good

**Summary:**

The authors propose a method for multi-person pose estimation that builds off Sparse R-CNN with additional functionality to achieve accurate pose estimates. In Sparse R-CNN, there are reference boxes/queries that are iteratively updated to converge on a final set of detections. In this work, there are additionally query embeddings (Q_p) that are decoded to produce keypoint estimates. The key contributions of this work are the “Spatial Part Embedding Generation Module” (SPEGM) and the “Selective Iteration Module” (SIM) which are used to update Q_p.

- SPEGM: Given a bounding box for a detected person, produce ROI aligned features from the box and use convolutions to predict a set of attention maps (each map corresponding to different body parts). Use these attention maps to perform a weighted pooling of the ROI aligned features to yield a set of embeddings E_p.
- SIM: Given the current embeddings, predict weights to perform a weighted sum of Q_p and E_p providing a new Q_p which is then decoded for the current pose estimate.

This process is repeated multiple times as the bounding boxes and part queries are iteratively improved. When benchmarking on COCO this method outperforms prior end-to-end methods.



**Questions:**

- For the SIM, instead of the proposed weighted sum were any standard recurrent updates considered (e.g. a GRU update)?

**Limitations:**

no discussion included by authors

**Strengths And Weaknesses:**

Strengths:

- There are a number of different pieces that come together in the proposed method, and the authors provide ablations showing that they all play a role (using part queries rather than predicting keypoints directly from the instance-level features, the SPEGM, the weighted sum for the SIM)
- The results are strong, outperforming prior work on very competitive benchmarks notably without using test-time augmentation (TTA)

Weaknesses:

- This method seems as though it must be fairly expensive, the authors report splitting a batchsize of 16 across 8 A100s, it would be helpful to see comparisons in terms of memory use and training and inference time compared to other methods.
- While many ablations are provided to show the benefits of the proposed approach, it is worth pointing out that some simpler baselines are close in their performance (e.g. a simple summation instead of the SIM (Table 2a), or a dynamic MLP instead of the SPEGM (Table 1)). I don’t know if the most compelling case has been made yet that the full pipeline in all its complexity is necessary. It feels as though a simpler set of layers and update rule could replace the SPEGM and SIM achieving the same level of performance. This criticism is perhaps a bit unfair, as the authors do clearly show that performance would be slightly worse with the simpler alternatives.
- The communication and organization of the paper could be improved. The method itself is interesting, but it can be challenging to work through some of the unwieldy names and acronyms. Also, I found the overview figure (Fig. 2) difficult to make sense of.

Overall:

Altogether I think the method is interesting enough and the benchmark results speak for themselves to lean towards acceptance, but the quality and organization of writing could be improved to better communicate the proposed ideas and justify the full pipeline.

Post rebuttal: Having read the other reviews and author response I lean towards accepting and raise my score.

---

> ### Author Response · Authors · 2022-08-01
> **Response to Reviewer 2Ux2**
>
> Thank you for the careful reviews and constructive suggestions. We clarify the questions as follows.
>
> **Q1:This method seems as though it must be fairly expensive, the authors report splitting a batchsize of 16 across 8 A100s, it would be helpful to see comparisons in terms of memory use and training and inference time compared to other methods.**
>
> **A1**: Thanks for your concern. Following the previous query-based methods (e.g., DETR, Deformable-DETR, and Sparse RCNN), the batch size of QueryPose is set to 16 without adjustment. Batch size 16 is enough to obtain superior results, while the state-of-the-art dense competitors (e.g., DEKR, SWAHR, AdaptivePose, LQR) often set batch size as 64 or even larger.  For memory use, QueryPose costs 14G GPU memory with ResNet50 and 23G with HRNet-W48. The above dense competitors cost over 30G GPU memory with HRNet-W48. We observe that the main cost is caused by dynamic MLP (a single dynamic MLP contains 11.3M parameters). The proposed SPEGM and SIM only contains 1.1M and 0.62M parameters. This phenomenon inspires us to reduce the complexity of dynamic MLP module later.
>
> We measure the training and inference times of QueryPose with different backbones.
> Backbone | AP| Training Time [h]| Inference Time [ms]
> --|--|--|--
> ResNet50| 68.7| 42| 70
> HRNet-W32|72.4|58| 100
> HRNet-W48|73.6| 62| 105
> Swin-L | 73.3| 67| 117
>
> Compared with the representative dense competitors, we observe QueryPose is training-efficient with fewer training epochs. Furthermore, QueryPose eliminates time-consuming post-processes and achieves the faster inference speed. The training and inference times are measured on the same device if possible. The detailed comparisons are reported as follows:
> Methods | AP| Training Epoch| Training Time [h]| Inference Time [ms]
> --|--|--|--|--
> HrHRNet-W48 [16]| 69.8 |300 | >140h | 317
> SWAHR-W48 [39]|70.8| 300 | >140h| 319
> DEKR-W48 [24]| 71.0|140 | >80h| 284
> AdaptivePose-W48 [20]| 70.0|280 | 140h| 110
> LQR-W48 [40]| 72.4| 170 | 100h| 297
> QueryPose-W48| 73.6| 68 | 62h| 105
> We also provide runtime performance in the revision.
>
> **Q2: While many ablations are provided to show the benefits of the proposed approach, it is worth pointing out that some simpler baselines are close in their performance (e.g. a simple summation instead of the SIM (Table 2a), or a dynamic MLP instead of the SPEGM (Table 1). I don’t know if the most compelling case has been made yet that the full pipeline in all its complexity is necessary. It feels as though a simpler set of layers and update rule could replace the SPEGM and SIM achieving the same level of performance. This criticism is perhaps a bit unfair, as the authors do clearly show that performance would be slightly worse with the simpler alternatives.**
>
> **A2**: Thanks. As reported in Table 1, first, we verify the superiority of part query over instance query across different feature interaction methods. Based on part query, we further validate the effectiveness of different feature interaction methods. Compared with dynamic MLP module with 11.3M parameters, we argue that SPEGM is more effective and efficient with only 1.1M parameters. It also reduces the training time and memory cost significantly with better performance. Furthermore, based on SPEGM, we verified different iteration schemes as shown in Table 2(a), SIM can improve the non-iteration and simple summation by 1.1 and 0.7 AP with 0.62M parameters, which proves SIM can adaptively boost the informative features and filter the noises with slight computational cost. Therefore, the proposed SPEGM and SIM are simple and efficient than other alternatives.
>
> **Q3: The communication and organization of the paper could be improved. The method itself is interesting, but it can be challenging to work through some of the unwieldy names and acronyms. Also, i found the overview figure (Fig. 2) difficult to make sense of.**
>
> **A3**:  Thanks for your valuable comments. We explicitly define and explain the unwieldy names and acronyms in the revision. Moreover, we will augment the communication and re-organize the paper writing for clearer understanding. Figure 2 will be further improved for more intuitive  illustration.
>
> **Q4: For the SIM, instead of the proposed weighted sum were any standard recurrent updates considered (e.g. a GRU update)?**
>
> **A4**: Thanks for your comments. We consider that the core insight of the proposed SIM is similar to GRU. Both of them leverage the update gate and reset gate to enhance the informative feature and filter the noise. SIM can be regard a simplified version of gated recurrent unit. We also attempt to use more complex structures in SIM, but it does not bring additional improvements.

---

> > ### Author Response · Authors · 2022-08-09
> > **Looking forward to hearing the response from reviewer 2Ux2**
> >
> > Thank you for the previous insightful suggestions. We also would like to hear your further comments. If you have any questions, please let us know.

---

### Official Review · Reviewer_f8E1 · 2022-07-18

**Rating:** 6
**Confidence:** 4
**Soundness:** 3 good
**Presentation:** 3 good
**Contribution:** 3 good

**Summary:**

This paper proposed an end-to-end multi-person pose regression framework to address the problem of high complexity and redundant post-processes existing in the previous work. To achieve this, the authors introduced the Spatial Part Embedding Generation Module (SPEGM) and the Selective Iteration Module (SIM) to encode human instances by a set of learnable part-level queries together with instance-level queries. Different from previous works, the part-level queries can capture the spatial details and structural information of human poses. Experimental results show that the proposed method achieved the state-of-the-art performance on MS COCO and CrowdPose datasets.

**Questions:**

See "Weaknesses".

**Limitations:**

Yes.

**Strengths And Weaknesses:**

__Strengths__

1. This paper is overall well-structured and easy-to-follow.

2. I think the problem this paper tackled is important for the community. Multi-person pose estimation is definitely an important problem as it is prohibitively expensive to collect ground truth with great human efforts for in-the-wild images.

3. The proposed method is technically sound and novel.

4. The results show that the performance of the proposed method is strong and even on par with two-stage methods.

__Weaknesses__

1. The authors suggest in the abstract and introduction of the paper that one of the main benefits of the proposed method is to reduce the complexity of the inference. However, there is no evidence to support this point. Detailed comparisons of runtime performance such as FLOPs and inference time should be reported.

2. The box detection pipeline can be replaced by a DETR-like architecture using object queries to further reduce the complexity of the model. I wonder if the authors have considered this design. If so, what’s the performance and why the current FPN-like architecture is preferred?

3. It is interesting to see in the supplementary that the proposed method outperforms two-stage models on CrowdPose by a quite notable margin. I’d like to see a more detailed justification/discussion on the results.

4. Some comments to improve the clarity of the paper: 1. In Figure 2, it is unclear to see how the results from the previous stage are used as the input of the current stage. 2. The results on CrowdPose are important and should be put into the main paper.

---

> ### Author Response · Authors · 2022-08-01
> **Response to Reviewer f8E1**
>
>
> **Q1: The authors suggest in the abstract and introduction of the paper that one of the main benefits of the proposed method is to reduce the complexity of the inference. However, there is no evidence to support this point. Detailed comparisons of runtime performance such as FLOPs and inference time should be reported.**
>
> **A1**: Thanks for your valuable comments. The existing dense end-to-end methods always require the complex hand-crafted post-processes (e.g., NMS, grouping or refinements) to obtain the final results. We observe that the complex post-processes always consumes more time than network forward in existing methods. QueryPose eliminates the time-consuming post-processes and achieves a compact inference pipeline.
>
> We report the accuracy, inference time and FLOPs of QueryPose with ResNet50, HRNet-W32, HRNet-W48 and Swin-L.
> | Backbone|  AP |  Time [ms] |  FLOPs  |
> |---- |----- |--- |- |
> |ResNet50      | 68.7  |    70   | 153G |
> |HRNet-W32  | 72.4 |    100 | 237G    |
> |HRNet-W48  | 73.6 | 105 |     334G   |
> |Swin-L |73.3 |117 | 545G   |
>
> Compared with the dense end-to-end methods, QueryPose can achieve faster inference speed. For example, the inference time of QueryPose-R50 is 70ms with 153G FLOPs, which is faster than Mask RCNN-R50 (114ms) with 238G FLOPs. We further list the comparisons for inference time with the most competitive dense end-to-end methods.  With the same backbone, QueryPose outperforms HrHRNet, SWAHR, DEKR, and LQR  in terms of accuracy and speed. The detailed comparisons of runtime performance are reported in the revision.
> | Methods |  Backbone  | AP| Time [ms] |
> |---|-- |--|---|
> |HrHRNet [16]  | HRNet-W48 |69.8 |317    |
> | SWAHR [39] |   HRNet-W48| 70.8 | 339  |
> |DEKR [24]  | HRNet-W48|71.0 | 284|
> |LQR [40] | HRNet-W48|72.4| 297 |
> | QueryPose| HRNet-W48 |73.6  | 105  |
>
> **Q2: The box detection pipeline can be replaced by a DETR-like architecture using object queries to further reduce the complexity of the model. I wonder if the authors have considered this design. If so, what’s the performance and why the current FPN-like architecture is preferred?**
>
> **A2**: Thanks for your insightful comments. Actually, the box decoder can be replaced by any DETR-like architecture. However, in the algorithm design process, we only concentrate on the keypoint decoder and aim to build a sparse end-to-end MPPE pipeline. Compared with DETR-like methods, Sparse RCNN avoids the cascade encoder and only interacts with the features of local ROIs, thus seems more simple in term of architecture. However, we observe a single dynamic MLP module used in Sparse RCNN is computationally expensive with 11.3M parameters. We will consider designing a more light module to replace dynamic MLP or adopting DETR-like architecture to reduce the network complexity in future work. We suppose the DETR-like architecture could still achieve equal performance.
>
> **Q3: It is interesting to see in the supplementary that the proposed method outperforms two-stage models on CrowdPose by a quite notable margin. I’d like to see a more detailed justification/discussion on the results.**
>
> **A3**: Thanks. Two-stage top-down approaches use two independent models including a human detector and a single-person pose estimator.  The reasons that the proposed method outperforms two-stage models on CrowdPose can be summarized in two aspects:
> + The reported two-stage methods generally use the dense human detector, which require the NMS to suppress the duplicates. However, in crowd scenes, the heavily overlapped instances may be removed after NMS, thus leading to inferior results. The phenomenon also can be observed in Table 11 of Sparse RCNN [27]. The sparse paradigm maybe more suitable for crowded scenes.
> + The single-person pose estimator leverages the heatmap to independently represent the keypoint position, while losing the keypoint relations. In crowded scenes, the cropped single person image always contains the limbs of other persons. Learning the keypoint associations can avoid confusion.
>
> QueryPose leverages sparse paradigm to solve the MPPE and employ the multi-head self-attention to capture the relationships across different part queries, thus is more robust for crowded scenes. We also provide the discussions in Appendix.
>
> **Q4: Some comments to improve the clarity of the paper: 1. In Figure 2, it is unclear to see how the results from the previous stage are used as the input of the current stage. 2. The results on CrowdPose should be put into the main paper.**
>
> **A4**: Thanks for your valuable comments. We leverage the predicted box of previous stage as the proposal box of current stage. The keypoint coordinates are predicted independently for each stage. We will further improve Figure 2 to explicitly illustrate in the revision. Moreover, the results on CrowdPose will be moved into the main paper for comprehensive presentation.

---

### Comment · Area_Chair_MG3p · 2022-08-03
**Author-Reviewer Discussion Period (Aug 2 - Aug 9)**

Dear Reviewers,

First of all, thank you for your service for the NeurIPS 2022 review.

The author-reviewer discussion period has just begun (from August 2nd to August 9th).

Please kindly take a look at all other reviews and the author responses (if any).

If you have any concerns that require more clarification from the authors, now it is the time to engage the discussion with the authors.

Since the discussion period is only one week, please properly exploit the chance.

Thanks,

---

> ### Comment · Area_Chair_MG3p · 2022-08-07
> **Please engage in the discussion with the authors**
>
> Dear Reviewers,
>
> This is a friendly reminder that there are only a few days left for the author-reviewer discussion (due August 9th).
>
> Please properly exploit the chance to engage in the discussion with the authors.
>
> Even a few short sentences to acknowledge that you have read the rebuttals and other reviewer comments are appreciated.
>
> Finally, the area chair thanks the reviewer f8E1 for acknowledging all their comments are addressed by the authors.
>
> Thanks,

---

### Meta-Review · Area_Chair_MG3p · 2022-08-23

**Recommendation:** Accept
**Confidence:** Certain

**Metareview:**

The authors propose a novel framework for end-to-end multi-person pose estimation by employing a set of learnable part-level queries along with instance-level queries. Promising results are demonstrated on the challenging COCO and CrowdPose datasets. The provided author rebuttal successfully addressed all reviewer concerns. As a result, all four reviewers recommend accepting the papers. The AC has read the paper, reviewer comments, author rebuttal, and all the discussions. The AC agrees with the reviewer recommendations. The authors are encouraged to include the rebuttal results (e.g., runtime analysis) to their camera-ready.

**Award:**

No

---

### Decision · Program_Chairs · 2022-09-14

Accept